# Multiblock Analysis Applied to TD-NMR of Butters and Related Products

**Jean-Michel Roger** [1,2,*] , **Silvia Mas Garcia** [1,2], **Mireille Cambert** [3] **and Corinne Rondeau-Mouro** [3]

1    INRAE, ITAP, Montpellier SupAgro, University of Montpellier, F-34096 Montpellier, France; silvia.mas-garcia@inrae.fr
2    ChemHouse Research Group, F-34096 Montpellier, France
3    INRAE, UR OPAALE, 17 Avenue de Cucillé, CS 64427, F-35044 Rennes, France; mireille.cambert@inrae.fr (M.C.); corinne.rondeau@inrae.fr (C.R.-M.)
*    Correspondence: jean-michel.roger@inrae.fr

**Abstract:** This work presents a novel and rapid approach to predict fat content in butter products based on nuclear magnetic resonance longitudinal ($T_1$) relaxation measurements and multi-block chemometric methods. The potential of using simultaneously liquid ($T_{1L}$) and solid phase ($T_{1S}$) signals of fifty samples of margarine, butter and concentrated fat by Sequential and Orthogonalized Partial Least Squares (SO-PLS) and Sequential and Orthogonalized Selective Covariance Selection (SO-CovSel) methods was investigated. The two signals ($T_{1L}$ and $T_{1S}$) were also used separately with PLS and CovSel regressions. The models were compared in term of prediction errors (RMSEP) and repeatability error ($\sigma_{rep}$). The results obtained from liquid phase (RMSEP $\approx$ 1.33% and $\sigma_{rep} \approx$ 0.73%) are better than those obtained with solid phase (RMSEP $\approx$ 5.27% and $\sigma_{rep} \approx$ 0.69%). Multiblock methodologies present better performance (RMSEP $\approx$ 1.00% and $\sigma_{rep} \approx$ 0.47%) and illustrate their power in the quantitative analysis of butter products. Moreover, SO-Covsel results allow for proposing a measurement protocol based on a limited number of NMR acquisitions, which opens a new way to quantify fat content in butter products with reduced analysis times.

**Keywords:** TD-NMR; T1 relaxation; chemometrics; multiblock; SO-PLS; SO-Covsel; butter

## 1. Introduction

The rapid characterization of agri-food products is an important industrial issue that is involved in different steps of processes such as control of raw materials, control of manufacturing processes and quality control of finished products. The use of non-destructive and rapid techniques is necessary to integrate this characterization into production workflows. Spectroscopic methods have developed in response to this need—e.g., those based on infrared, Raman, terahertz and nuclear magnetic resonance. These techniques have advantages and disadvantages in terms of cost, speed of acquisition and sensitivity to the desired property. While infrared spectrometry is largely used in several domains for its sensitivity and its speed, Nuclear Magnetic Resonance (NMR) has many applications in food science, due to its good resolution in mixtures of multi-phasic samples. However, the main disadvantage of NMR is its weak sensitivity with long acquisition times that make the study of food processes very time consuming. Any development that can speed up the NMR measurement protocol is thus welcome.

Among the NMR techniques, time domain nuclear magnetic resonance (TD-NMR) spectroscopy is particularly useful for the characterization of fat in food products [1–4]. TD-NMR proceeds by applying to a sample placed in a magnet a sequence of radiofrequency (RF) fields to excite the magnetic moments of the nuclei and record their return to equilibrium after the radiofrequency switch off. From

this process, a decaying signal called the Free Induction Decay (FID) is obtained. Other dedicated sequences yield the acquisition of the two intrinsic parameters in NMR, namely longitudinal relaxation time ($T_1$) and transverse relaxation time ($T_2$). $T_2$ is widely used to measure solid fat content and the water and oil amounts of foodstuffs, based on well-known international standard methods (AOCS Cd 16b-93, AOCS Cd 16-81, ISO 8292, IUPAC 2.150). However, these NMR standard methods are often non-applicable to non-anhydrous products 1. Moreover, there is little application of these methods in complex food systems characterized by a phase-compositional diversity of molecules, such as butters and related products. In these systems, the interpretation of the signal can be very tedious since chemical and diffusional proton exchanges modulate the signal intensities.

In terms of signal analysis, chemometrics offers efficient solutions to process TD-NMR data. In the 1990s, Rutledge et al. [5,6] showed that chemometrics offers fast and efficient methods, such as partial least-squares regression (PLS) [7], for determining the information content of the signals. Demonstrations on margarine and butters for determining their water content, or the water mobility in starchy-lignin samples were rather convincing. Since then, PLS were mainly used on TD-NMR data for the rapid evaluation of oil or water content in food samples [8–13].

Although PLS is still the most popular chemometric tool, other methods can be useful to assist in the processing of TD-NMR data. Thus, chemometrics offers methods for merging several measurements performed on the same samples, referred to as multiblock analysis methods. [14]. Various multiblock PLS methods have been developed following similar rationales [15]. More recently the Sequential and Orthogonalized PLS (SO-PLS) was proposed [16]. This method uses a successive decomposition of the blocks in order to take advantage of the full predictive potential of the data. Chemometrics also provides methods for variable selection. The purpose of variable selection is twofold: First, identify the parts of the signal that are most important for characterizing a product. This type of selection can be performed by examining the loadings of factorial methods, such as PLS or Principal Component Analysis (PCA), by using iterative methods such as iPLS [17], or by using criteria such as Variable Importance in Projection (VIP) [18]. The second objective of variable selection is to identify a small number of variables on which a calibration model can be built. This could be of great interest in NMR to reduce analysis time. The most commonly used methods in chemometrics with this purpose are the sparse methods [19], or variance based methods such as SPA [20] or CovSel [21]. The variable selection in a multiblock calibration framework has been little investigated. The VIP concept has been adapted to SO-PLS [22], which allows for detecting important areas in the signals as in classical PLS. The combination of SO-PLS and CovSel recently led to SO-CovSel [23], which allows for selecting a very small number of variables from multiblock data, most related to a predicted response.

In this work, we proposed to apply the multiblock analysis methods on NMR measurements in complex foodstuffs. As mentioned before, most of the standard NMR methods use transverse $T_2$ relaxation time measurements or combination of both $T_1$ and $T_2$ measurements for water and/or fat quantification [2]. Here, we propose the use of $T_1$ measurements alone, to estimate the fat content in various samples (butters and related samples) on the basis of NMR signals describing the crystallized and liquid phases separately.

## 2. Material and Methods

### 2.1. Samples

Fifty commercial samples of butter, margarine and concentrated fat (anhydrous milk fat) were purchased from all over Europe. Their fat content in % *w/w* $F_w$ was determined using the EN ISO 3727 method. Table 1 summarizes the composition of the dataset.

**Table 1.** Dataset summary. $F_w$ is the % *w/w* of fat.

| Type | #Samples | Mean $F_w$ (%) | Min $F_w$ (%) | Max $F_w$ (%) |
|---|---|---|---|---|
| Butter | 35 | 82.5 | 81.5 | 84.9 |
| Margarine | 3 | 79.6 | 78.3 | 80.6 |
| Concentrated fat | 12 | 99.4 | 98.1 | 99.9 |

## 2.2. Methods

TD-NMR measurements were performed using a Minispec BRUKER spectrometer (Wissembourg, Germany) operating at a resonance frequency of 20 MHz. The system was equipped with a temperature control device connected to a calibrated optic fiber (Neoptix Inc., Québec, QC, Canada) allowing for ±0.1 °C temperature regulation. All the samples were measured in triplicate and at two temperatures: 6 °C and 15 °C.

The Fast Saturation Recovery sequence (FSR) was used [24]. It consisted of a 90° pulse, a delay $\tau_1$ and a last 90° pulse. The FID signals were recorded over 100 µs. One hundred values of $\tau_1$ were used, ranging from 5 to 4000 ms for measurements at 6 °C and from 5 to 5000 ms for measurements at 15 °C. The intensities of two points of the FID signal were recorded for each $\tau_1$ value: the first one ($I_{1T}$) at 11 µs and the second one ($I_{1L}$) at 70 µs. These two signals are related to the total matter and the liquid matter, respectively. The variation of these two signals according to $\tau_1$ yielded two signals, $T_{1T}$ and $T_{1L}$, as illustrated on Figure 1. The signal related to the solid matter was determined by: $T_{1S} = T_{1T} - T_{1L}$.

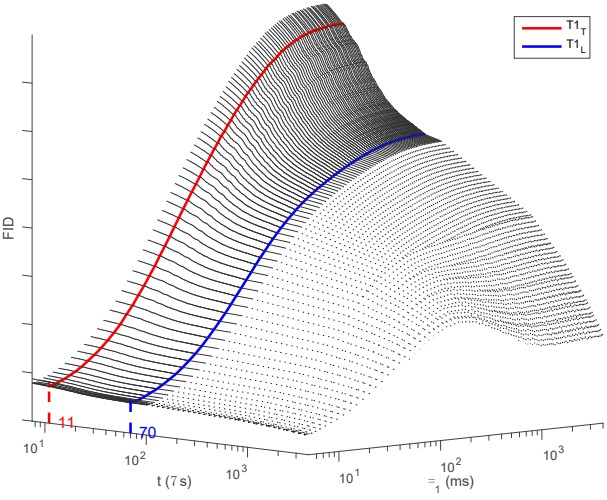

**Figure 1.** Scheme of both signal measurements: FID (in light gray) and the signal actually measured (in dark grey).

## 2.3. Data Processing

$T_{1L}$ and $T_{1S}$ signals were imported into Matlab and used to build-up the **X** ($N \times Q$) matrices of descriptors (data matrix $\mathbf{X_L}$ and $\mathbf{X}$s, respectively), where $N$ corresponds to the number of samples and $Q$ to the number of delays $\tau_1$. Fat contents of the samples were used as the **y** vector of responses ($N \times 1$).

### 2.3.1. PLS and SO-PLS

PLS aims at finding links between a matrix of descriptors **X** ($N \times Q$) and a matrix of responses **Y** ($N \times R$). It decomposes the two matrices into a sum of $K$ rank 1 products of scores by loadings, as:

$$\mathbf{X} = \mathbf{TP}^T + \mathbf{R}_X \quad \text{and} \quad \mathbf{Y} = \mathbf{UQ}^T + \mathbf{R}_Y, \tag{1}$$

with the aim of covariance between columns of **T** and of **U** being maximal and columns of **T** being mutually orthogonal. The PLS is usually followed by a classical linear regression between **T** and **Y**, yielding a PLSR. The scores used in the regression step are called latent variables. When the **Y** matrix contains only one column, PLSR yields an estimate $\hat{\mathbf{y}}$ of **y** according to:

$$\hat{\mathbf{y}} = \mathbf{Xb} \tag{2}$$

The vector **b**, called b-coefficients, is a vector of dimension $Q$, as the variables contained in the rows of **X**. Consequently, it could be plotted as a signal. Interpretation of its shapes helps to analyze the relative importance of the **X** features in the regression. For example, a positive peak in the b coefficients corresponds to an area where the signal is positively related to **y**. Another example is when all signals have similar shapes, b coefficients are shaped like the average signal. However, some more complex characteristics can also be analyzed using b coefficients, such as slopes, curvatures and even horizontal offsets, as explained in [25].

SO-PLS is a multi-block regression method where the information is sequentially extracted from the different predictor blocks. Considering the case of two predictor blocks ($\mathbf{X}_1$ and $\mathbf{X}_2$) used to estimate a **y** response, the algorithm is as follows:

1. The **y** response is fitted to $\mathbf{X}_1$ by PLS using $K_1$ latent variables.
2. $\mathbf{X}_2$ is orthogonalized with respect to the $K_1$ scores extracted from the first block.
3. The **y**-residual obtained from step 1 is fitted to $\mathbf{X}_2$ by PLS using $K_2$ latent variables.
4. The scores produced at steps 1 and 3 are then used in an MLR to predict the response **y**.

A SO-PLS model contains as many b-coefficients as blocks used by the model.

### 2.3.2. CovSel and SO-CovSel

CovSel 21 aims at selecting in **X** the $K$ most effective variables to predict **y**. This algorithm proceeds by $K$ iterations of the following steps:

1. Calculation of the squared covariance between the columns of **X** and **y**:

$$\mathbf{c} = diag(\mathbf{X}^\mathrm{T}\mathbf{y}\mathbf{y}^\mathrm{T}\mathbf{X}) \tag{3}$$

2. Selection of the variable $I$ corresponding to the maximum value of **c**:

$$I = ArgMax(c_i) \tag{4}$$

3. Projection of **X** and **y** orthogonally to the column $I$ of **X**:

$$\mathbf{X} = \mathbf{X} - (\mathbf{x}_I(\mathbf{x}_I^\mathrm{T}\mathbf{x}_I)^{-1}\mathbf{x}_I^\mathrm{T})\mathbf{X} \tag{5}$$

$$\mathbf{y} = \mathbf{y} - (\mathbf{x}_I(\mathbf{x}_I^\mathrm{T}\mathbf{x}_I)^{-1}\mathbf{x}_I^\mathrm{T})\mathbf{y} \tag{6}$$

Similar to the PLSR, a Multiple Linear Regression (MLR) is calculated between the selected variables of **X** and **y**.

SO-CovSel is similar to SO-PLS, where PLS is replaced by CovSel. Consequently, the SO-CovSel algorithm is the same as the one explained above for the SO-PLS where latent variables are replaced by selected variables.

### 2.3.3. Figures of Merit

Two data subsets were prepared, one to calibrate the model (with 37 samples) and another one to externally validate the previous calibration model (with 13 samples). The 50 samples were sorted in

ascending order of $F_w$. One in four samples were placed into the test set and the rest into the calibration set, providing a similar distribution of $Fw$ in both sets. Since all the samples were measured in triplicate, the calibration set contained $N_c = 37 \times 3 = 111$ signals and the test set $N_t = 13 \times 3 = 39$ signals.

PLS, CovSel, SO-PLS and SO-CovSel were applied on the two blocks $\mathbf{X_L}$ and $\mathbf{X_S}$ to predict $\mathbf{y}$, with $K$, $K_1$ and $K_2$ varying from 0 to 10. A one-sample-out cross-validation provided an estimate $\hat{\mathbf{y}}$ of the $N_c$ calibration samples for each $K$ or pair $(K_1, K_2)$. The RMSECV cross-validation error was calculated using the following formula:

$$RMSECV(K) = \sqrt{\frac{\sum_i (y_i - \hat{y}_i(K))^2}{N_c}} \text{ for PLS and CovSel} \tag{7}$$

$$RMSECV(K_1, K_2) = \sqrt{\frac{\sum_i (y_i - \hat{y}_i(K_1, K_2))^2}{N_c}} \text{ for SO-PLS and SO-CovSel} \tag{8}$$

The model was calculated on the whole calibration set and then reapplied to this set, which provided an estimate $\hat{\mathbf{y}}$ of the $N_c$ values of $F_w$ for each $K$ or pair $(K_1, K_2)$. The RMSEC calibration error was calculated according to the following formula:

$$RMSEC(K) = \sqrt{\frac{\sum_i (y_i - \hat{y}_i(K))^2}{N_c - K - 1}} \text{ for PLS and CovSel} \tag{9}$$

$$RMSEC(K_1, K_2) = \sqrt{\frac{\sum_i (y_i - \hat{y}_i(K_1, K_2))^2}{N_c - K_1 - K_2 - 1}} \text{ for SO-PLS and SO-CovSel} \tag{10}$$

Examination of the evolution of RMSEC and RMSECV values as a function of $K$ or $(K_1, K_2)$ made it possible to choose the optimal numbers of latent variables $K_{opt}$ or $(K_{1opt}, K_{2opt})$. A model was then calibrated with these values on the whole calibration set and was applied to the test set, providing an estimate $\hat{y}$ of the $N_t$ fat values. The following figures of merit were calculated:

$$\text{RMSEP} = \sqrt{\frac{\sum_i (y_i - \hat{y}_i)^2}{N_t}} \quad Bias = \frac{\sum_i (y_i - \hat{y}_i)}{N_t} \quad SEP = \sqrt{\frac{\sum_i (y_i - \hat{y}_i - Bias)^2}{N_t}} \tag{11}$$

All the data processing was carried out similarly on the signals acquired at 6 °C and at 15 °C. All the data processing was made using Matlab (The Mathworks Inc., Natick, MA, USA). Matlab code of SO-PLS and SO-Covsel can be freely downloaded from: https://www.chem.uniroma1.it/romechemometrics/research/algorithms/.

## 3. Results and Discussion

### 3.1. NMR Signal Description

Figure 2 shows the mean signals for the three classes of product: butter, concentrated fat and margarine, for both sets of $T_1$ relaxation data, $T_{1L}$ and $T_{1S}$, and at both temperatures. All the signals have the same sigmoidal appearance. It can be noticed that, by increasing the temperature, the intensity of the $T_{1L}$ signals increases, while the intensity of the $T_{1S}$ signals decreases. This observation is coherent with the solid fat content of samples that is expected to decrease at higher temperatures due to fat crystal melting. The solid fat phase should be characterized by $T_{1S}$ whose intensity should decrease at 15 °C. As no material is lost during the NMR measurements, the solid signal decreases in favor of the liquid one, therefore the $T_{1L}$ signal intensity increases.

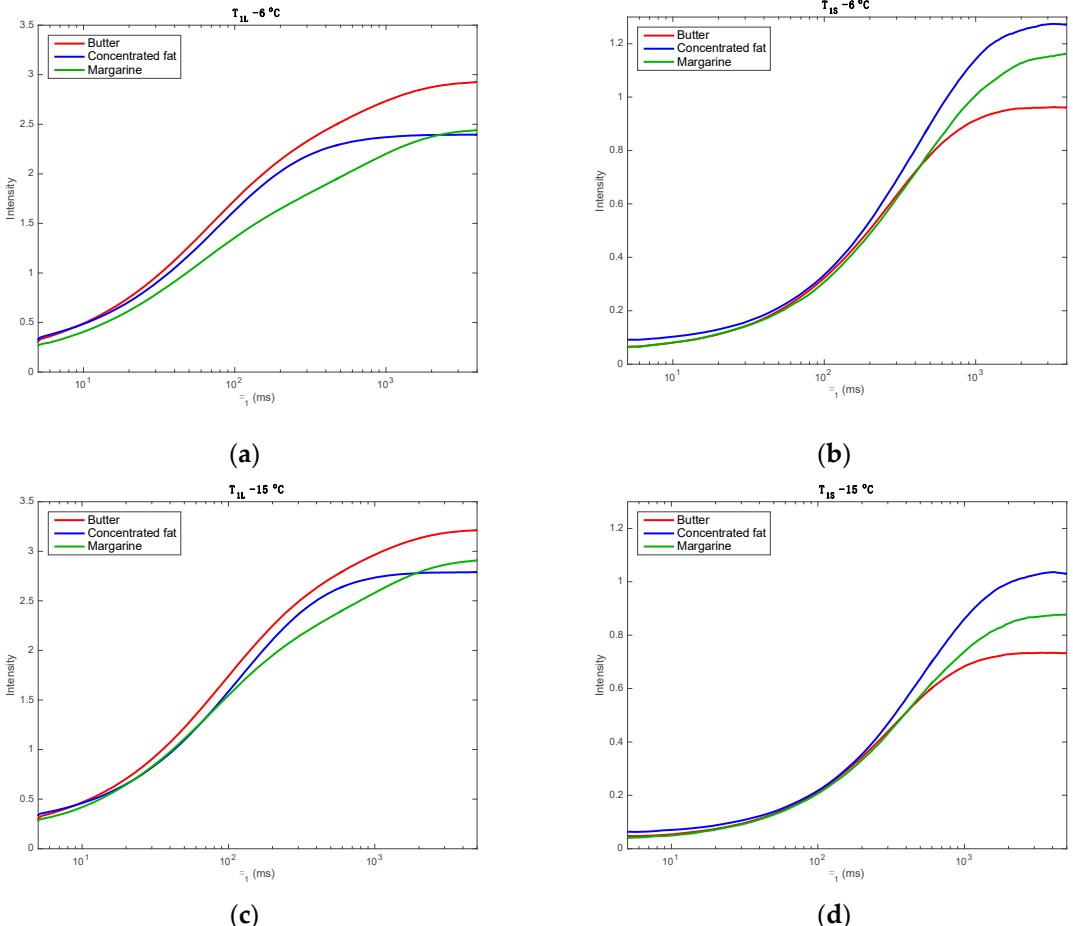

**Figure 2.** Mean signal of each class of products. (**a**,**c**): $T_{1L}$, (**b**,**d**): $T_{1S}$, (**a**,**b**): T = 6 °C, (**c**,**d**): T = 15 °C. Vertical scale of (**b**,**d**) is different from that of (**a**,**c**). Horizontal scale is logarithmic.

Regarding $T_{1L}$, it can be noticed that the margarine signals (in green) are below the butter signals (in red), and globally below the concentrated fat signals (in blue). This discrimination is clearer at 6 °C than at 15 °C. At high times (>3000 ms), the signal intensity of butters is the highest one, while for margarines the signal intensity is slightly lower, showing a clear increase with higher temperature. For both temperatures, the concentrated fat signal is the only one that presents a complete sigmoid with a clear plateau reached after 1000 ms. Butter signals also seem to be like a sigmoid with a plateau after 3000 ms. The margarine signals have a more crushed shape between 100 and 1000 ms and no plateau is reached for $T_{1L}$. However, the signal intensity for margarines exceeds that of the concentrated fat signal after 1000 ms, and particularly at 15 °C. The crushed shape of the signal of margarines indicates the presence of proton pools with different $T_{1L}$ compared to the other butters. These protons may characterize emulsifiers or additives used in the margarine composition. They are, in general, lecithin or synthetic molecules that crystallize at 6 °C [26]. By increasing the temperature at 15 °C, the intensity of the margarine mean signal increases. This can be explained by the appearance of a new proton phase that should originate from the melting of these additives.

Regarding $T_{1S}$, the change in temperature does not induce any change in the form or order of the signals. The signals measured at 6 °C simply appear to be vertically expanded compared to those measured at 15 °C. The changes due to the increasing of temperature are only due to the proton quantity that only influences the signal intensity. However, expected changes in the signal shape due to the variations in relaxation time $T_{1S}$ were not observed. The signals of the three classes are very similar for times before 200 ms. For longer times, the signal intensities for butter are lower than for margarines, which in turn are lower compared to concentrated fats. This result agrees with the solid fat content

of concentrated fats. These samples are expected to be exclusively composed of fat. Margarines are expected to contain a higher content of dry matter compared to butters. As mentioned before, they contain added emulsifying agents that provide their special texture and which crystallized, in general, at 6 °C [26]. The fact that the $T_{1S}$ signal for margarines is higher than butters can be explained by this additional crystalline phase.

### 3.2. Mono Block Calibrations

Table 2 reports the figures of merit of the models. PLS and CovSel models were calibrated and tested on $T_{1L}$ and $T_{1S}$ successively, and SO-PLS and SO-CovSel on $T_{1L}$ and $T_{1S}$ jointly.

**Table 2.** Figures of merit of the models at the two different temperatures. $R_x^2$ are the square correlation coefficients between actual and predicted values, $\sigma_{rep}$ is the standard deviation of repeatability calculated on the triplicates. #LV is the number of variables used in the models.

| | | T (°C) | #LV $T_{1L}$ $T_{1S}$ | RMSEC (%) | $R_c^2$ | RMSECV (%) | $R_{cv}^2$ | RMSEP (%) | SEP(%) | Bias (%) | $\sigma_{rep}$ (%) |
|---|---|---|---|---|---|---|---|---|---|---|---|
| $T_{1L}$ | PLS | 6 | 6 | 1.05 | 0.98 | 1.33 | 0.97 | 1.15 | 1.14 | −0.18 | 0.55 |
| | | 15 | 7 | 1.12 | 0.98 | 1.32 | 0.97 | 1.29 | 1.24 | −0.36 | 0.61 |
| | CovSel | 6 | 7 | 1.00 | 0.98 | 1.33 | 0.97 | 1.22 | 1.20 | −0.21 | 0.63 |
| | | 15 | 8 | 1.11 | 0.98 | 1.32 | 0.97 | 1.33 | 1.28 | −0.37 | 0.73 |
| $T_{1S}$ | PLS | 6 | 2 | 4.60 | 0.61 | 5.01 | 0.54 | 4.61 | 4.50 | 1.01 | 0.60 |
| | | 15 | 2 | 3.91 | 0.71 | 4.31 | 0.66 | 5.20 | 5.20 | 0.22 | 0.67 |
| | CovSel | 6 | 2 | 4.64 | 0.60 | 5.03 | 0.53 | 4.60 | 4.53 | 0.83 | 0.61 |
| | | 15 | 2 | 3.93 | 0.71 | 4.30 | 0.66 | 5.27 | 5.27 | 0.17 | 0.69 |
| SO−PLS | | 6 | 61 | 0.77 | 1.00 | 0.96 | 0.99 | 0.68 | 0.68 | −0.04 | 0.32 |
| | | 15 | 72 | 0.81 | 0.99 | 1.11 | 0.99 | 1.00 | 0.95 | −0.32 | 0.47 |
| SO−CovSel | | 6 | 71 | 0.79 | 0.99 | 1.05 | 0.99 | 0.77 | 0.77 | −0.08 | 0.41 |
| | | 15 | 81 | 0.89 | 0.99 | 1.16 | 0.99 | 0.92 | 0.89 | −0.23 | 0.45 |

Regardless of the signals and methods used, the RMSEP is often lower than the RMSECV, which may seem surprising. PCA analyses on the four calibration sets of spectra, using enough components to explain 99.99% of the variance, were carried out. The presence of some extreme points was revealed by a test on the scores [27]. These extreme points may affect the cross-validation statistics but not the predictive capability of the model.

It can be seen that CovSel-based methods require more variables than those based on PLS for $T_{1L}$ data and the same number for $T_{1S}$. This could be expected due to the fact that CovSel is a degraded PLS. However, the performance of CovSel-based methods is very close to that of the PLS-based methods. It can also be noted that models at 15 °C generally perform worse than models at 6 °C. This is due to the fact that at 6 °C the crystalline phase is more important, giving a higher proportion of the $T_{1S}$ signal and hence an improvement in the model performance.

The use of $T_{1S}$ alone gives poor results with standard errors 3–5% and $R^2$ barely exceeding 0.7. It can be seen from the RMSECV curves (data not shown) that the error decreases slightly at $K = 2$ and then increases sharply, showing that the model fails to fit. The use of $T_{1L}$ alone gives much better results, with RMSECs slightly above 1%, RMSECVs of around 1.3% and RMSEPs of around 1.2%. The bias is always less than 0.5% in absolute value and the repeatability error is between 0.5 and 0.7%. It is worth mentioning that the $T_{1L}$ signal corresponds to about the 60 and 85% of the total signal compared to the signal of $T_{1S}$ and, therefore, a better performance on $T_{1L}$ signals could be explained.

### 3.3. Multi Block Calibrations

Table 2 shows that better results are achieved using $T_{1L}$ and $T_{1S}$ simultaneously. All standard errors are less than 1%. The best results are obtained using the SO-PLS model at 6 °C, which presents a SEP of 0.68% and a bias almost null. This result is very satisfactory compared to those reported in the literature using the T2 signal for the fat content estimation: 3.1% in [28], from 2.7% to 4.4% in [13], from 0.45% to 0.84% in [9] and 1.2% in [8]. The repeatability error is 0.32%, which is almost two times lower than the repeatability error achieved using the standard Soxhlet method in [29]. The SO-CovSel model

at 6 °C also gave good results. The SO-PLS model at 15 °C is much worse than at 6 °C, with a SEP of 0.95%, a bias of −0.32% and a repeatability error of 0.47%. The SO-CovSel at 15 °C performs slightly better, with a SEP of 0.89%, a bias of −0.23% and a repeatability error of 0.45%.

It is remarkable to mention that calibration models using $T_{1T}$ ($T_{1T} = T_{1L} + T_{1S}$) were carried out and gave errors of about 4% (data not shown). Therefore, the use of SO-PLS and SO-CovSel combining the two signals, $T_{1L}$ and $T_{1S}$, in a fusion process presents a good alternative to quantify fat content in butter-related products.

Figure 3 shows the values of *Fw* predicted by the different multiblock models as a function of the *Fw* values measured by the reference method. Figure 3a shows the great results of SO-PLS at 6 °C. It can be seen that each of the three replicates is well predicted. Only one replicate of the margarine class is significantly underestimated. In Figure 3b it can be observed that the predictions of SO-CovSel at 6 °C present approximately the same behavior. In addition to the margarine sample, already poorly predicted by SO-PLS, a group of three replicates of a concentrated fat was underestimated. In Figure 3c the effect of dispersion's increase in the SO-PLS predictions, mainly for concentrated fat, is observed. On the other hand, it can be seen in Figure 3d that, in this example, SO-Covsel is much more robust to temperature change than SO-PLS.

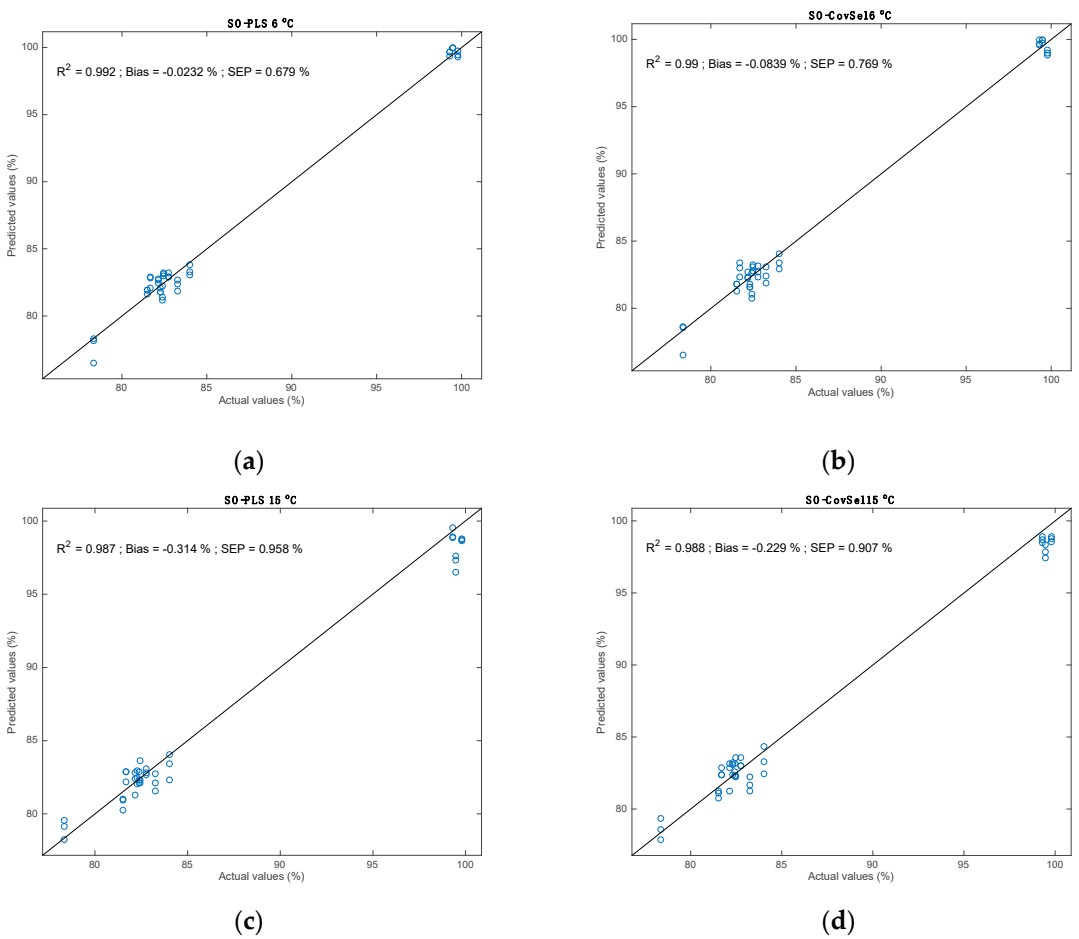

**Figure 3.** Predictions of $F_w$ on the test set by SO-PLS (**a**,**c**) and by SO-CovSel (**b**,**d**) for 6 °C (**a**,**b**) and for 15 °C (**c**,**d**).

Figure 4 shows the b-coefficients of the SO-PLS models superimposed on the mean spectra of the three product classes. These coefficients are convoluted with the $T_{1L}$ and $T_{1S}$ signals to calculate an estimate of $F_w$. Thus, the analysis of their shape allows a better understanding of which parts of the signal are useful for calibration, and how they are used by the model.

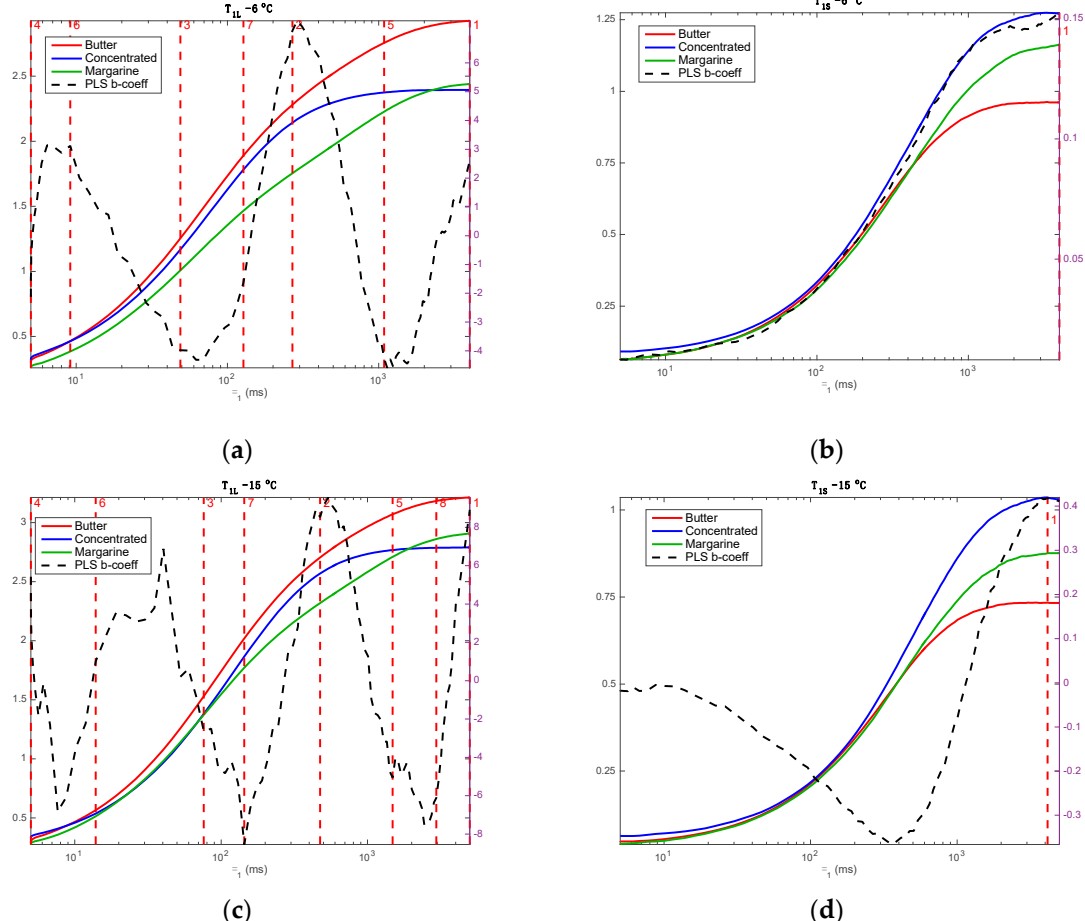

**Figure 4.** SO-PLS b-coefficients and variables selected by SO-CovSel, superimposed to the product mean spectra. Left (**a**,**c**): For $T_{1L}$ signals. Right (**b**,**d**): For $T_{1S}$ signals. Top (**a**,**b**): T = 6 °C. Bottom (**c**,**d**): T = 15 °C. Horizontal scale is logarithmic. Vertical dashed red lines show the relaxation times selected by CovSel.

Figure 4a shows the SO-PLS b-coefficients of $T_{1L}$ signal acquired at 6 °C. These calibration coefficients show two positive peaks, at 8 and 300 ms, and two negative peaks, at 65 and 1100 ms. Except for the peak at 8 ms, the coefficients values are very close to the $T_{1L}$ relaxation time values determined by the Inverse Laplace Transform of the $T_{1L}$ signal (data not shown). The b-coefficients for $T_{1S}$ (Figure 4b), do not show any peak. They have the same shape as the average $T_{1S}$ signal. They contribute to the estimation of $F_w$ by calculating the overall signal strength.

Figure 4 also shows the relaxation times selected by SO-CovSel. As a reminder, a multiple linear regression of the signals measured at these times provides an estimate of *Fw*, according to the performances reported in Table 2. On the $T_{1L}$ signal (Figure 4a), the variables selected, in decreasing order of importance, are: 4000, 269, 49, 5, 1035, 10 and 128 ms. All these variables are located at times such as overall height, peak height and/or depth, or sigmoid curvature, that allow the model to take into account the overall shape of the signal. In the $T_{1S}$ signal (Figure 4b), only one variable is selected, at 4000 ms. It allows the model to take into account the global intensity of the $T_{1S}$ signal.

Figure 4c shows the SO-PLS b-coefficients of the $T_{1L}$ signal acquired at 15 °C. These coefficients show two positive peaks at 30 and 400 ms and three negative peaks at 8, 150 and 2500 ms. For the $T_{1S}$ signal (Figure 4d), the b-coefficients show a negative peak at 350 and a positive peak at 4000 ms. They allow the model to take into account the slope of the right-hand side of the signal. Variables selected by SO-CovSel on the $T_{1L}$ signal at 15 °C are shown in the same figure. The variables selected, in descending order of importance, are 5000, 474, 76, 5, 1553, 19, 207 and 2949 ms. As before, all these

variables are located close to the peaks of the SO-PLS b-coefficients. They allow for the calculation of signal shape parameters, such as the overall height, the height and/or depth of the peaks, or the curvature of the sigmoid. On the $T_{1S}$ signal (Figure 4d), only one variable is selected, at 4146 ms. It allows the model to take into account the overall intensity of the $T_{1S}$ signal. This intensity is related to the nature of the product (see Figure 2) and especially to the quantity of solid matter.

All these observations demonstrate that the best models use complex information in the $T_{1L}$ signal, related to its shape, and very simple information in the $T_{1S}$ signal, related to its intensity. The complex shape of the $T_{1L}$ signal, especially for margarines, is due to its complex composition with an aqueous phase rich in emulsifiers that does not exist in butters or concentrated fats. In contrast, the $T_{1S}$ is only modulated by the quantity of dry matter that is weakly higher in margarines and by the crystalline phase that changes with temperature.

## 4. Discussion

The traditional method of measuring fat in food products is the Soxhlet method, which is time consuming, destructive and uses polluting solvents. TD-NMR has been used to solve these problems. International standards have even been established (AOCS Cd 16b-93, AOCS Cd 16-81, ISO 8292, IUPAC 2.150). However, these NMR standard methods are often non-applicable to non-anhydrous products 1. This paper shows that the use of multi-block methods combining solid and liquid phase TD-NMR signals makes it possible to measure the fat content of fatty products containing water.

TD-NMR signals are produced after one or several radiofrequency pulses, followed by relaxation phenomena that result in the exponential decay of the signals with time constants characteristic of the molecule mobility in the material being analyzed. Consequently, the classical signal processing in TD-NMR data consists of least squares fitting procedures [30] or uses a numerical inversion of the Laplace transform [31]. Such a signal processing task can be rather fastidious and is known to be an ill-conditioned and ill-posed problem, resulting in a large number of solutions that small noise in the data can easily affect. Chemometric methods have been developed to process ill-conditioned data in a stable and robust manner. Their application to process TD-NMR raw data is presently demonstrated as a good solution to overcome the above methodological limitations.

The results provided by CovSel, with mono-block or multi-block data, show that it is possible to considerably simplify and improve the standards measurement protocols, avoiding the destruction of the sample and the use of pollutant solvents in comparison to standard Soxhlet methods and using the TD-NMR raw data directly without further fastidious preprocessing methods. For both temperatures, $T_{1L}$ measurements can be performed based on eight or even less sequences, which is an excellent timesaver. With eight measurements and a recycle time of 5000 ms, the experiment should last only 40 s, which is three to four times less than what is needed for $T_2$ relaxation time measurements with 16 scans and an echo time of 0.1 ms.

The results reported in this article are as good as or better than those obtained by existing methods. In particular, the repeatability error is excellent, which shows the stability and robustness of the chemometric methods used. The results are good for both measurement temperatures. This suggests that it would be possible to build models independent of temperature, which would greatly shorten and simplify the measurement protocol, making it even more suitable for online use. Chemometrics offers a lot of methods for building such robust models.

## 5. Conclusions

The strategy based on nuclear magnetic resonance longitudinal ($T_1$) relaxation measurements and multi-block chemometric methods has been shown to be useful for an accurate estimation of the fat content in complex products. This work also demonstrates that chemometrics methods offer alternative approaches to analyze the raw TD-RMN data without fastidious preprocessing methods. It is essential to stress the benefits of using, in a combined way, information from liquid ($T_{1L}$) and solid ($T_{1S}$) phases of butters and related samples. These data were obtained by a skilled acquisition of the

FID intensities at 11 and 70 µs. On the one hand, the results obtained from SO-PLS and SO-CovSel improved those obtained from classical methods. Moreover, SO-CovSel results allow for proposing a measurement protocol based on a limited number of measurements, which reduce the analysis time.

Further research is ongoing to test this variable fusion and selection approach on other combinations of TD-NMR signals, such as $T_1$ and $T_2$.

**Author Contributions:** J.-M.R. was in charge of carrying out the chemometrics processing, writing the main parts of the article, especially focusing on the chemometrics parts, S.M.G. was in charge of discussing the results and collaborating in writing the chemometrics parts, M.C. was in charge of setting the experiment protocols and of carrying out the RMN measurements, C.R.-M. was involved in the experimental protocol design, in interpreting the RMN measurements and in writing the parts dedicated to RMN. All authors have read and agreed to the published version of the manuscript.

**Funding:** This research received no external funding.

**Acknowledgments:** The NMR experiments were performed using the NMR facilities of the AgroScans Platform (PRISM, Rennes, France).

**Conflicts of Interest:** The authors declare no conflict of interest.

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
