# Peer review of "Multiblock Analysis Applied to TD-NMR of Butters and Related Products"

_applsci, doi:10.3390/app10155317_

Round 1

Reviewer 1 Report

the study is correctly carried out and the work is well presented

Author Response

Changes in the new version are in red.

Reviewer's comment:

the study is correctly carried out and the work is well presented

Response :

Thank you

Reviewer 2 Report

The manuscript focuses on the application of multi block regression modelling combined by variable selection approaches to NMR data from solid and liquid phase for the rapid prediction of fat content in three different and most commercially available fatty foodstuff. The main advantage of proposed methodology is the possibility of saving time of analysis thanks to the identification of the most predictive and meaningful variables related to fatty content, thus suggesting the potential to be also applicable in a real, practical scenario. In addition, the work topic is of current interest in the field of chemometrics applied to food science, where research has been continuously moving towards the high throughput melting of information in order to increase accuracy of prediction and overall performances.  

The abstract is well designed and clearly summarize the main objectives of the work as well as the most meaningful results achieved. Similarly, all the sections of the manuscript are well organized from a stylistic point of view and enclose what is supposed to be read. A general suggestion is to improve the quality of the introduction section since, in the first part, it is too unbalanced toward the theoretical aspect of the NMR. I expected to find a focus on the meaning of the application of the methodology to the chosen foodstuffs. Why butter, margarine and concentrated fats were chosen? Along similar lines, why NMR was chosen over other more rapid and cheaper techniques (e.g. FT-IR, NIR spectroscopies) which are well known to be used for the same purpose? A little focus relative to the comparison with these techniques could be of interest. Similarly, the conclusion parts of the manuscript could also benefit from remarking the practical usefulness of the proposed methodology that goes beyond the time saving, e.g. easy implementation? Higher accuracy? Transferability of the methodology?

  • The revision of some typos and formatting error throughout the whole manuscript is necessary since hider the correct interpretation of the text, for instance:
    • All figures and tables are not properly identified with number, but with an error message.
    • The absence of brackets for numbered references in the text is misleading.

  • Reference number 1 is missing since the reference list has slipped by one.

  • Please reformulate the sentence reported at page 2, lines 62–63 y correcting grammar and style. 

  • The verbal form “build” at line 69 must be changed with “built”.

  • Please, change “time analysis” with “analysis time” (line 70).

  • Methods, Section 2.3.3., Figures of Merit (page 4). Please specify if selection was made manually or by applying some selection algorithm. Moreover, I see that 3 out 9 samples replicas of margarines were included in the validation set, thus ensuring a 1:3 partition for these samples. If the same partition was respected also for butter (n=35*3) and concentrated fats (n=12*3), I expected a total of 3+35+12 samples to be included into the validation set, but, in the present work, the validation sample (replicas) were 39. Why different partition rules were chosen for each sample group (margarine, butters, and fats)? Please, provide further information.
  • Figure 2 (caption). Please, correct the typo T=06 °C removing 0.

  • In Mono block calibration section (page 6, lines 213–216) the presence of outliers is supposed to be responsible for higher RMSECV values compared to RMSEP values and, therefore, for hindering cross-validation statistics. This statement should be more strongly supported by investigating the effective presence of anomalous samples/outliers through the application of proper tool for identifying them (e.g. Hotelling’s T2 statistics). This does not mean that outliers should be removed and the models recomputed after elimination, but they are suggested to be properly identified and not only supposed to be present.

  • Page 7, lines 217–222. The authors write “It can be seen that CovSel methods systematically require more variables than those based on PLS"– please, specify that this information only refers to T1L dataset, since the number of variables for T1S dataset is the same (i.e 2 variables) for CovSel and PLS, as reported in Table 2.

Author Response

Changes in the new version are in red.

It seems that the word document deposited to the journal website has encountered some formatting problems. Some bookmarks to figures, tables and references were lost, that made the reading difficult. We apologize for that. In the new version, all the bookmarks have been replaced by normal text in order to avoid the problem.

Reviewer's comment:

The manuscript focuses on the application of multi block regression modelling combined by variable selection approaches to NMR data from solid and liquid phase for the rapid prediction of fat content in three different and most commercially available fatty foodstuff. The main advantage of proposed methodology is the possibility of saving time of analysis thanks to the identification of the most predictive and meaningful variables related to fatty content, thus suggesting the potential to be also applicable in a real, practical scenario. In addition, the work topic is of current interest in the field of chemometrics applied to food science, where research has been continuously moving towards the high throughput melting of information in order to increase accuracy of prediction and overall performances. 

The abstract is well designed and clearly summarize the main objectives of the work as well as the most meaningful results achieved. Similarly, all the sections of the manuscript are well organized from a stylistic point of view and enclose what is supposed to be read.

A general suggestion is to improve the quality of the introduction section since, in the first part, it is too unbalanced toward the theoretical aspect of the NMR. I expected to find a focus on the meaning of the application of the methodology to the chosen foodstuffs. Why butter, margarine and concentrated fats were chosen?

Response:

We agree with the reviewer.The introduction has been rewritten according to the reviewer's suggestions. Some text has been added at the beginning to introduce the context and some parts related to the RMN theory have been suppressed.

Reviewer's comment:

Along similar lines, why NMR was chosen over other more rapid and cheaper techniques (e.g. FT-IR, NIR spectroscopies) which are well known to be used for the same purpose? A little focus relative to the comparison with these techniques could be of interest.

Response:

The aim of the paper was ti investigate the use of chemometrics on NMR signals. It was not to provide the best solution for fat characterization. Some sentences have been added in the introduction about NMR vs other techniques.

Reviewer's comment:

Similarly, the conclusion parts of the manuscript could also benefit from remarking the practical usefulness of the proposed methodology that goes beyond the time saving, e.g. easy implementation? Higher accuracy? Transferability of the methodology?

Response:

A discussion part has been added in the new version.

Reviewer's comment:

The revision of some typos and formatting error throughout the whole manuscript is necessary since hider the correct interpretation of the text, for instance:

All figures and tables are not properly identified with number, but with an error message.

Response:

It seems that the word document deposited to the journal website has encountered some formatting problems. Some bookmarks to figures, tables and references were lost, that made the reading difficult. We apologize for that. In the new version, all the bookmarks have been replaced by normal text in order to avoid the problem. Moreover, we will deposit a pdf which will not be affected by any formatting problem.

Reviewer's comment:

The absence of brackets for numbered references in the text is misleading.

Response:

This absence of brackets is also due to a difference of Word style between our version and the MDPI’s one. We apologize for that.  In the new version, the references are between brackets.

Reviewer's comment:

Reference number 1 is missing since the reference list has slipped by one.

Response:

One more time, this problem was due to a formatting problem. It is solved in the new version.

Reviewer's comment:

Please reformulate the sentence reported at page 2, lines 62–63 y correcting grammar and style.

Response:

Done

Reviewer's comment:

The verbal form “build” at line 69 must be changed with “built”.

Response:

Done

Reviewer's comment:

Please, change “time analysis” with “analysis time” (line 70).

Response:

Done

Reviewer's comment:

Methods, Section 2.3.3., Figures of Merit (page 4). Please specify if selection was made manually or by applying some selection algorithm. Moreover, I see that 3 out 9 samples replicas of margarines were included in the validation set, thus ensuring a 1:3 partition for these samples. If the same partition was respected also for butter (n=35*3) and concentrated fats (n=12*3), I expected a total of 3+35+12 samples to be included into the validation set, but, in the present work, the validation sample (replicas) were 39. Why different partition rules were chosen for each sample group (margarine, butters, and fats)? Please, provide further information.

Response:

We agree that this point needs further information.

The splitting between calibration and validation sets was made on the basis of the fat content (Fw) only. The samples were sorted in ascending order of Fw. Then, one sample on four was put in the validation set and the rest in the calibration set. By this way, the distributions of Fw for the two sets were similar. Some text has been added in the new version to explain this procedure.

Reviewer's comment:

Figure 2 (caption). Please, correct the typo T=06 °C removing 0.

Response:

Done

Reviewer's comment:

In Mono block calibration section (page 6, lines 213–216) the presence of outliers is supposed to be responsible for higher RMSECV values compared to RMSEP values and, therefore, for hindering cross-validation statistics. This statement should be more strongly supported by investigating the effective presence of anomalous samples/outliers through the application of proper tool for identifying them (e.g. Hotelling’s T2 statistics). This does not mean that outliers should be removed and the models recomputed after elimination, but they are suggested to be properly identified and not only supposed to be present.

Response:

Following your suggestion, we investigated the actual presence of samples that might appear as outliers in the cross-validation process. We have used a test on T2 and Q limits of a PCA , with enough PC to explain 99.99% of variance. The text has been modified accordingly.

Reviewer's comment:

Page 7, lines 217–222. The authors write “It can be seen that CovSel methods systematically require more variables than those based on PLS"– please, specify that this information only refers to T1L dataset, since the number of variables for T1S dataset is the same (i.e 2 variables) for CovSel and PLS, as reported in Table 2.

Response:

Correction has been done in the text.

Reviewer 3 Report

The work must be thoroughly revised. The received version of the work looks like it was a draft, did the authors read it before they submitted it to the journal? The biggest complaint concerns the lack of discussion. The problematic is also justification of undertaking research in the context of the current methods of fat analysis in the analyzed products. Will the new methods be faster, cheaper, more accurate, because generally speaking the practical meaning of the analyzes will not be widely used due to the lack of availability of equipment? After these corrections the paper can be revised again

Author Response

Changes in the new version are in red.

Reviewer's comment:

The work must be thoroughly revised. The received version of the work looks like it was a draft, did the authors read it before they submitted it to the journal? The biggest complaint concerns the lack of discussion. The problematic is also justification of undertaking research in the context of the current methods of fat analysis in the analyzed products. Will the new methods be faster, cheaper, more accurate, because generally speaking the practical meaning of the analyzes will not be widely used due to the lack of availability of equipment? After these corrections the paper can be revised again

Response:

We agree that there were many problems with this document, which made it difficult to read. We apologize for the inconvenience caused. The new version has been thoroughly revised. The introduction section has been completely rewritten and a discussion section has been added.

We hope that this new version will meet your expectations.

Round 2

Reviewer 3 Report

Now it can be published